# Breaking the Symmetry: Resolving Symmetry Ambiguities in Equivariant Neural Networks

**Sidhika Balachandar**                                    SIDHIKAB@CS.CORNELL.EDU
*Cornell University, Ithaca, NY, USA*

**Adrien Poulenard**                                       PADRIEN@STANFORD.EDU
**Congyue Deng**                                           CONGYUE@STANFORD.EDU
**Leonidas Guibas**                                        GUIBAS@CS.STANFORD.EDU
*Stanford University, Stanford, CA, USA*

**Editors:** Sophia Sanborn, Christian Shewmake, Simone Azeglio, Arianna Di Bernardo, Nina Miolane

## Abstract

Equivariant networks have been adopted in many 3-D learning areas. Here we identify a fundamental limitation of these networks: their ambiguity to symmetries. Equivariant networks cannot complete symmetry-dependent tasks like segmenting a left-right symmetric object into its left and right sides. We tackle this problem by adding components that resolve symmetry ambiguities while preserving rotational equivariance. We present OAVNN: Orientation Aware Vector Neuron Network, an extension of the Vector Neuron Network (Deng et al., 2021). OAVNN is a rotation equivariant network that is robust to planar symmetric inputs. Our network consists of three key components. 1) We introduce an algorithm to calculate symmetry detecting features. 2) We create a symmetry-sensitive orientation aware linear layer. 3) We construct an attention mechanism that relates directional information across points. We evaluate the network using left-right segmentation and find that the network quickly obtains accurate segmentations. We hope this work motivates investigations on the expressivity of equivariant networks on symmetric objects.
**Keywords:** O(3) equivariance, planar symmetry, 3-D learning, point cloud analysis

## 1. Introduction

3-D representations of real-life objects are used for problems in computer vision, robotics, medicine, augmented reality, and virtual reality. Geometric deep learning leverages the geometric properties of 3-D structures to build robust and data-efficient models for tasks in these fields. Many networks have been proposed for 3-D geometric learning and point cloud analysis, as seen in (Guo et al., 2021). For tasks on 3-D data, the input has no prevailing pose, and the network must perform well regardless of the input pose. These tasks lend themselves to invariance and equivariance. Invariant models produce the same output regardless of the input pose. Equivariant models produce outputs that are transformed in the same way as the input. Thus these tasks motivate rotation equivariant and invariant networks that share information across all rotated poses. These networks have shown better performance and data efficiency for unaligned data on tasks such as segmentation and reconstruction in (Deng et al., 2021); shape retrieval and scene classification in (Esteves et al., 2019); and 3-D model recognition and atomization energy regression in (Cohen et al., 2018).

Figure 1: The VNN has ambiguities to symmetries and cannot complete symmetry-dependent tasks. We created the OAVNN that is rotation equivariant and robust to planar symmetries. Consider the symmetry-dependent task of left-right segmentation of left-right symmetric objects. The VNN cannot complete this task, but the OAVNN learns the correct segmentation.

Nevertheless, as shown in Figure 1, many of these networks have issues with symmetry. Equivariant networks are equivariant to symmetries. Therefore these networks cannot complete tasks that depend on the input symmetry. Due to the prevalence of symmetric inputs and symmetry dependent tasks, there is a need for an equivariant network that handles symmetries. We introduce OAVNN: Orientation Aware Vector Neuron Network, Figure 2, an extension of the Vector Neuron Network (VNN) (Deng et al., 2021). The OAVNN is a rotation equivariant network that is robust to planar symmetric inputs. The code for the model is available at https://github.com/sidhikabalachandar/oavnn.

## 2. Related Work

Many equivariant designs are summarized in (Esteves, 2020). One category uses spherical harmonics, such as (Thomas et al., 2018) and (Weiler et al., 2018). The networks in (Cohen et al., 2019) and (Esteves et al., 2019) are similar but instead are equivariant to the symmetry group of an icosahedron. A simpler formulation is the Vector Neuron Network (Deng et al., 2021) which is based on interpretable 3-D vectors. Here, we focus on simple, interpretable networks like the VNN which has symmetry issues that we intend to study and improve.

A related computer graphics problem is symmetry detection, as seen in (Gao et al., 2021) and (Shi et al., 2020). In this project, we not only study symmetry detection but also resolve expressivity issues of equivariant networks related to symmetries.

## 3. A Fundamental Problem in Equivariant Networks

### 3.1. Definitions

We consider the group O(3) of all rotation and reflection matrices. A function $f$ is O(3) **equivariant** if for all $\boldsymbol{R} \in \mathrm{O}(3)$ and $\boldsymbol{x} \in \mathbb{R}^3$, $f(\boldsymbol{R} \cdot \boldsymbol{x}) = \boldsymbol{R} \cdot f(\boldsymbol{x})$, and O(3) **invariant** if $f(\boldsymbol{R} \cdot \boldsymbol{x}) = f(\boldsymbol{x})$. An O(3) equivariant function can be made invariant by taking the norm.

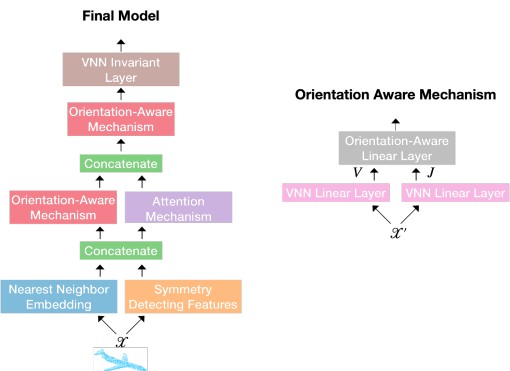

Figure 2: **Framework** — The OAVNN takes input $\mathcal{X}$ and passes it through the nearest neighbor embedding defined in Section 3.3 to create local patches. Additionally, as defined in Section A.1 the symmetry detecting features are extracted. These features are passed through the attention mechanism defined in Section A.3 and the orientation aware mechanism (right), which uses the orientation aware linear layer defined in Section A.2. The outputs are passed through another orientation aware mechanism. For tasks like segmentation, a final invariant layer is applied.

### 3.2. Ambiguity under Self-Symmetries

Equivariant networks display problems in encoding objects with self-symmetries. Consider object $\mathcal{X} \in \mathbb{R}^{C \times 3}$ with self-symmetry $\boldsymbol{R}_0$. When encoded by an equivariant $f$, due to the self-symmetry, $f(\boldsymbol{R}_0\mathcal{X}) = f(\mathcal{X})$. Additionally, since $f$ is equivariant, $f(\boldsymbol{R}_0\mathcal{X}) = \boldsymbol{R}_0 f(\mathcal{X})$. Thus, $f(\mathcal{X}) = \boldsymbol{R}_0 f(\mathcal{X})$. Let's call $f(\mathcal{X})_\perp$ the component of $f(\mathcal{X})$ in the **direction of symmetry** or the component orthogonal to $\boldsymbol{R}_0$'s plane of symmetry. Then $f(\mathcal{X})_\perp = \boldsymbol{R}_0 f(\mathcal{X})_\perp = -f(\mathcal{X})_\perp$. Thus we see that $f(\mathcal{X})_\perp$ must be zero. *Therefore for a planar symmetric input, an* $\mathrm{O}(3)$ *equivariant function cannot predict information in the direction of symmetry.* This means an equivariant network will have trouble solving tasks that distinguish between two symmetric parts, such as symmetric segmentation.

### 3.3. Vector Neuron Network: An Example

Here we focus on a specific equivariant network, the VNN (Deng et al., 2021). The VNN's linear layer is structured as $f_{lin}(\mathcal{X}) = \mathcal{W}\mathcal{X}$, where $\mathcal{X} \in \mathbb{R}^{C \times 3}$ is a list of input vectors and $\mathcal{W} \in \mathbb{R}^{C' \times C}$ is a learnable weight matrix. The VNN maintains $\mathrm{O}(3)$ equivariance since it treats each vector input as an independent unit, and a matrix $\boldsymbol{R} \in \mathrm{O}(3)$ commutes with this linear layer as $f_{lin}(\mathcal{X} \cdot \boldsymbol{R}) = \mathcal{W}\mathcal{X} \cdot \boldsymbol{R} = f_{lin}(\mathcal{X}) \cdot \boldsymbol{R}$. The VNN also modifies ReLU, pooling, and batch normalization layers. All modified layers operate on vectors and preserve equivariance. The VNN also proposes an invariant layer.

The VNN has two drawbacks. First, since it is $\mathrm{O}(3)$ equivariant, it cannot learn the direction of symmetry. Second, the VNN can only share information between local patches of points. The VNN creates local patches through a nearest neighbor embedding by concatenating each point with its $k$ nearest neighbors. For symmetry-dependent tasks, the

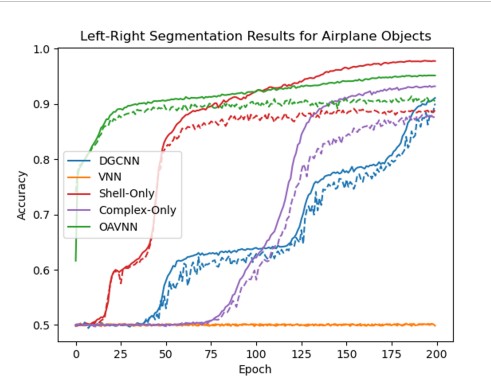

Figure 3: Results for the left-right segmentation experiment described in Section 5 for airplane objects. The plot shows training (solid) and testing accuracies (dashed) for 1) DGCNN (not equivariant), 2) VNN (equivariant and ambiguous to symmetries), 3) Shell-Only (symmetry-detecting features and attention mechanism), 4) Complex-Only (orientation aware linear layer and attention mechanism), and 5) OAVNN (symmetry-detecting features, orientation aware linear layer, and attention mechanism).

network must also globally transfer information. Consider a left-right symmetric object. The left-right direction of symmetry is uniquely determined by the top-bottom and front-back directions. When learning symmetries, these directions must be globally transferred. Since the VNN cannot globally propagate information it has difficulty learning symmetries.

## 4. Methods

We introduce an orientation aware framework (Figure 2) to resolve the limitations of the VNN for planar symmetric inputs. We modify the VNN by adding three components: symmetry-detecting features defined in Section A.1, an orientation aware linear layer defined in Section A.2, and an attention mechanism defined in Section A.3. Finally, we apply the VNN invariant layer for tasks such as segmentation.

## 5. Results

We test five models using a left-right segmentation experiment run on symmetric airplane objects from the Shapenet dataset (Chang et al., 2015). Descriptions of the models are in Section C and the results are shown in Figure 3. Since the segmentation is symmetric, the model must detect the direction of symmetry. The VNN is ambiguous to symmetries and therefore fails to segment the airplane. Any network sensitive to the direction of symmetry can complete the task. However, the DGCNN model (Wang et al., 2019), a conventional segmentation network which is not equivariant, takes a long time to learn. When we add symmetry-detecting features to create the Shell-Only network, we can complete the task faster. The OAVNN model, which combines both symmetry-detecting features and the

orientation aware linear layer, can obtain an accurate segmentation faster than any other model. Therefore, as more symmetry-detecting features are added, the model completes the task faster. A more extensive ablation study is provided in Appendix Section C. Experimental results on other object classes (caps and chairs) are also provided in Appendix Section C.

## 6. Conclusions

Overall we have shown that equivariant networks are equivariant to symmetries, so their outputs are ambiguous to symmetries. Thus these networks have trouble completing symmetry-dependent tasks. We propose the OAVNN model, which resolves symmetry ambiguities for planar symmetric objects while preserving rotational equivariance. In the future, we hope to address more complex symmetries and symmetry-dependent tasks. Overall, we hope this work motivates investigations into the symmetry ambiguities of equivariant networks.

## Acknowledgments

We thank the Stanford Undergraduate Research Internship in Computer Science (CURIS) program for partly funding this work.

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

## Appendix A. Methods

Here we provide more detailed explanations of the three orientation aware components of the OAVNN.

### A.1. Symmetry Detecting Hand-Crafted Features

We propose hand-crafted features that detect the direction of symmetry. We calculate these features by using the planar symmetry detection algorithm. The algorithm is described in Figure 4, and the pseudocode is in Algorithm 1. It takes in two inputs: the point cloud $\mathcal{X} \in \mathbb{R}^{C \times 3}$ and the number of shells $n$. We let $\mathcal{X}_i$ represents the $i$th point in the point cloud where $i \in \{0, \ldots, C-1\}$. For each point, we split the point cloud into $n$ distance-based shells. We let $\mathcal{S}_{i,j}$ represent each distance based shells where $j \in \{0, \ldots, n-1\}$. For each shell, we calculate the shell vector from the starting point $\mathcal{X}_i$ to the centroid of the shell. We let $\boldsymbol{v}_{i,j}$ represent the shell vector. We then calculate the $\frac{n(n-1)}{2}$ directed cross-products $\boldsymbol{c}_{i,j,k}$ between nearer and further shell vectors. $\boldsymbol{c}_i$ represents the cross vector for a point $i$ (average of all directed cross products). For some geometric intuition, the shell vectors increase in length and rotate, and the cross vector represents the axis of rotation. The algorithm returns the average cross vector $\boldsymbol{c}$ across all points. Symmetric points will have shell vectors rotating in opposite directions. Therefore many components of the cross vectors will cancel out, and for a symmetric input, the average cross vector lies along the direction of symmetry. A proof of this claim is in Appendix Section B. The symmetry detecting feature is the average cross vector which informs the network about the object's symmetries.

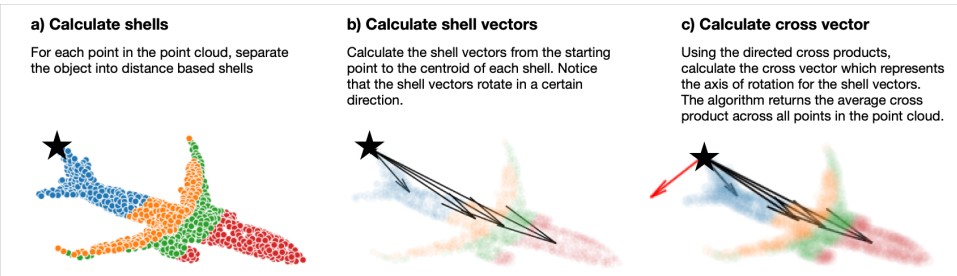

Figure 4: As shown in Algorithm 1, the planar symmetry detection algorithm identifies the direction of symmetry by calculating shell vectors (black) and cross vectors (red). It outputs the average cross vector across all points. The shell vectors for symmetric points, such as on the left and right wings, rotate in opposite directions. Therefore, many components cancel out, and in the end, the output lies in the direction of symmetry.

---

**Algorithm 1:** Pseduocode for the Planar Symmetry Detection Algorithm

---

**Input:** $\mathcal{X} \in \mathbb{R}^{C \times 3}, n$

**Output:** $c$, the direction of symmetry

**for** $i \leftarrow 0, C - 1$ **do**

    **for** $j \leftarrow 0, n - 1$ **do**

        $\mathcal{S}_{i,j} \leftarrow j\frac{C}{n}$ to $(j+1)\frac{C}{n}$ nearest neighbors

        $\boldsymbol{v}_{i,j} \leftarrow \mathbf{mean}(\mathcal{S}_{i,j} - \mathcal{X}_i)$

    **end**

    **for** $j \in [n]$ **do**

        **for** $k \in [j+1, n]$ **do**

            $\boldsymbol{c}_{i,j,k} \leftarrow \boldsymbol{v}_{i,j} \times \boldsymbol{v}_{i,k}$

        **end**

    **end**

    $\boldsymbol{c}_i \leftarrow \mathbf{mean}_{j,k}(\boldsymbol{c}_{i,j,k})$

**end**

$\boldsymbol{c} \leftarrow \mathbf{mean}_i(\boldsymbol{c}_i)$

---

### A.2. Orientation Aware Linear Layer

We also encourage detection of the symmetry direction by creating an orientation aware **complex linear layer** shown in Figure 5. The complex linear layer uses orientation to distinguish the direction of symmetry. It takes in a list of input vectors $\mathcal{V} \in \mathbb{R}^{N \times C \times 3}$ and a list of direction vectors $\mathcal{J} \in \mathbb{R}^{N \times C \times 3}$. Here $N$ represents the number of points in the point cloud, and $C$ represents the dimension of the channel that the linear layer acts on. The complex linear layer outputs a list of vectors in $\mathbb{R}^{N \times C' \times 3}$. Thus it changes the dimensionality of the channel from $C$ to $C'$. The complex linear layer learns a rotation and dilation for each vector in $\mathcal{V}$ in the direction of its corresponding vector in $\mathcal{J}$. It does this by defining two terms. First, it creates a list of $\mathbb{R}^3$ bases $\boldsymbol{R}(\mathcal{J}) = (\mathcal{U}_1, \mathcal{U}_2, \mathcal{J}) \in \mathbb{R}^{N \times C \times 3 \times 3}$

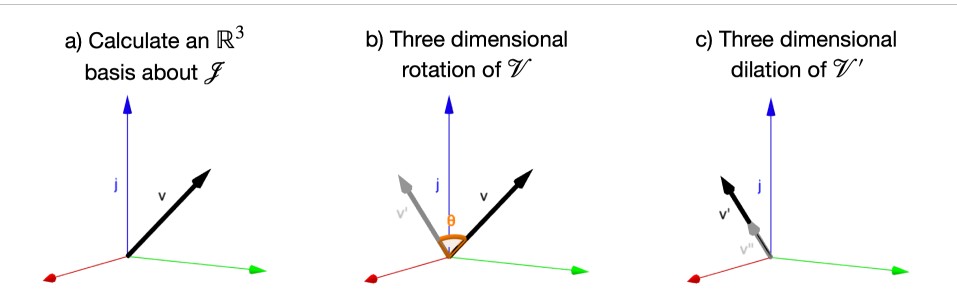

Figure 5: The orientation aware complex linear layer takes in a list of input vectors $\mathcal{V}$ and a list of direction vectors $\mathcal{J}$. It creates an $\mathbb{R}^3$ basis oriented about each $J$. Next, it rotates and dilates $\mathcal{V}$ within the basis. This layer learns the direction of symmetry because at least one of the basis vectors must have a nonzero component along the direction of symmetry. Additionally, this layer is sensitive to reflections because when $\mathcal{J}$ is negated, we get a rotation in the opposite direction.

where the list of the last vectors in each basis is $\mathcal{J}$. Each basis is positively oriented, and therefore for each basis $b$ in $\boldsymbol{R}(\mathcal{J})$, $\det b = +1$. The complex linear layer rotates and dilates each input vector in $\mathcal{V}$ in its corresponding basis in $\boldsymbol{R}(\mathcal{J})$ using a combined weight matrix

$$\mathcal{Z}(\mathcal{A}, \mathcal{B}, \mathcal{C})_{j,k,:,:} = \begin{bmatrix} \mathcal{A}_{j,k} & -\mathcal{B}_{j,k} & 0 \\ \mathcal{B}_{j,k} & \mathcal{A}_{j,k} & 0 \\ 0 & 0 & \mathcal{C}_{j,k} \end{bmatrix} \tag{1}$$

where $\mathcal{A}, \mathcal{B}, \mathcal{C} \in \mathbb{R}^{C' \times C}$ are learnable weight matrices and $j \in \{0, 1, \dots, C' - 1\}$ and $k \in \{0, 1, \dots, C - 1\}$. The complex linear layer outputs a list of vectors in $\mathbb{R}^{N \times C' \times 3}$ using the following formulation

$$f_{complex\_lin}(\mathcal{V}, \mathcal{J})_{i,j,:} = \sum_{k=0}^{C-1} \boldsymbol{R}(\mathcal{J})_{i,k,:,:} \mathcal{Z}(\mathcal{A}, \mathcal{B}, \mathcal{C})_{j,k,:,:} \boldsymbol{R}(\mathcal{J})_{i,k,:,:}^{\top} \mathcal{V}_{i,k,:} \tag{2}$$

where $j$ and $k$ are defined as before and $i \in \{0, 1, \dots, N - 1\}$. Thus $i$ iterates over the point dimension, $k$ iterates over the original number of channels, and $j$ iterates over the new number of channels.

The complex linear layer learns the direction of symmetry because at least one of the basis vectors must have a component in the direction of symmetry. Additionally, it is sensitive to symmetry reversal, because if $\mathcal{J}$ is flipped, we get a rotation in the opposite direction. This layer is inspired by Section 3.4 in DiffusionNet (Sharp et al., 2022), where $\mathcal{J}$ is the oriented normals to the surface. However, in the complex linear layer, $\mathcal{J}$ is learned.

### A.3. Attention Mechanism

We also create an attention mechanism to encourage global information transfer. Consider a left-right symmetric airplane. If we identify the up-down and front-back directions, we can

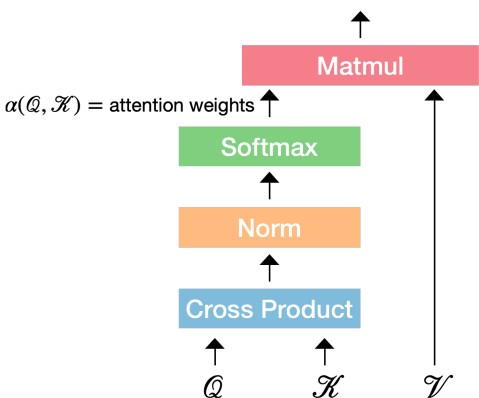

Figure 6: The attention mechanism takes in keys $\mathcal{K}$, queries $\mathcal{Q}$, and values $\mathcal{V}$. The goal of the attention mechanism is for each point to learn an oriented basis corresponding to the entire object. We assume that each point contains directional information. Each point then queries other points to learn orthogonal information. These orthogonal vectors are detected using the cross product operation. The original vectors combined with the orthogonal vectors create a local reference frame at each point.

uniquely determine the left-right direction. Since the airplane is approximately symmetric in the up-down direction, only the tail will inform us about this direction. We must propagate this information to all points.

To transfer information we propose an attention mechanism. Attention was first introduced in (Bahdanau et al., 2015), and it defines the relationship between keys, queries, and values (Vaswani et al., 2017). The attention mechanism creates a database of key and value pairs, and when given a query, it determines which pairs to attend to. Attention has been considered for equivariant networks on point clouds in works such as (Fuchs et al., 2020). However, we consider a different setting where each point has directional information. Then, at each point $\boldsymbol{x}$ we look for orthogonal information over all other points to create a local reference frame at $\boldsymbol{x}$. The orthogonal frame will be used by the complex linear layer which requires both a direction vector and an orthogonal direction to operate on. Therefore, we design our attention weights to be large when taking in a pair of query and key vectors that are orthogonal. Mathematically, for keys $\mathcal{K} \in \mathbb{R}^{C \times 3}$ and queries $\mathcal{Q} \in \mathbb{R}^{C \times 3}$, our attention weight formula is $\alpha(\mathcal{Q}, \mathcal{K})_{j,k} = \mathrm{softmax}_k \|\mathcal{Q}_{j,:} \times \mathcal{K}_{k,:}\|_2$ where $j \in \{0, 1, \ldots, C-1\}$ and $k \in \{0, 1, \ldots, C-1\}$. A diagram of the attention mechanism is shown in Figure 6.

## Appendix B. Proof of Planar Symmetry Detection Algorithm

**Theorem 1** *For a point cloud input that is symmetric about exactly one plane, the planar symmetry detection algorithm presented in Algorithm 1 outputs a vector orthogonal to the input object's plane of symmetry.*

**Proof** We will call the input point cloud $\mathcal{X}$. Without loss of generality, let $\mathcal{X}$ be symmetric about the $yz$-plane. We will show that the planar symmetry detection algorithm outputs a

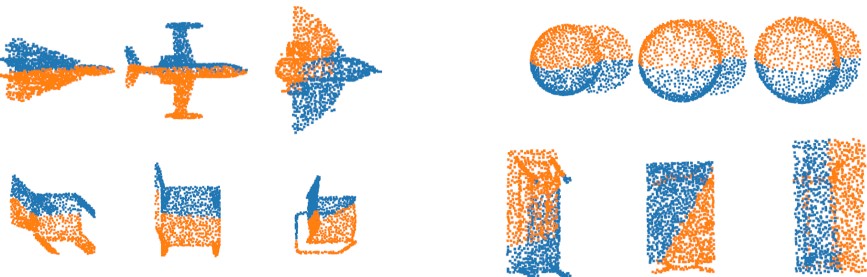

Figure 7: We conduct a left-right segmentation experiment on four different classes of objects from the Shapenet dataset (Chang et al., 2015) — airplanes (top-left), caps (top-right), chairs (bottom-left), and tables (bottom-right). These experiments only use the output of the planar symmetry detection algorithm. The algorithm returns a vector that represents the direction of symmetry. The plane which passes through the origin and is normal to the vector defines a classifier. The algorithm successfully segments objects with exactly one plane of symmetry, such as airplanes, caps, and chairs.

vector with only an $x$-component. For any point $\boldsymbol{x}$ in the point cloud, there exists a point $\boldsymbol{x}'$ which is symmetric to $\boldsymbol{x}$ about the $yz$-plane. Let $\boldsymbol{s}_1 = (x_1, y_1, z_1), \ldots, \boldsymbol{s}_n = (x_n, y_n, z_n)$ be the $n$ shell vectors of $\boldsymbol{x}$. Since $\boldsymbol{x}'$ is the reflection of $\boldsymbol{x}$ about the $yz$-plane, the shell vectors of $\boldsymbol{x}'$ will be the reflection of the shell vectors of $\boldsymbol{x}$ about the $yz$-plane. Therefore the shell vectors of $\boldsymbol{x}'$ are $\boldsymbol{s}'_1 = (-x_1, y_1, z_1), \ldots, \boldsymbol{s}'_n = (-x_n, y_n, z_n)$. The directed cross products of the shell vectors of $\boldsymbol{x}$ are of the form $(y_j z_k - y_k z_j, x_k z_j - x_j z_k, x_j y_k - x_k y_j)$ for all $j \in \{0, \ldots, n-1\}$ and $k \in \{0, \ldots, n-1\}$ where $k > j$. The directed cross products of the shell vectors of $x'$ are of the form $(y_j z_k - y_k z_j, -(x_k z_j - x_j z_k), -(x_j y_k - x_k y_j))$. For any fixed $(j, k)$, the sum of the directed cross product for $\boldsymbol{x}$ and $\boldsymbol{x}'$ will only have an $x$-component. Since the vector returned by the planar symmetry detection algorithm is the average of all directed cross products across all points, the output vector of the algorithm will also only have an $x$-component. Thus the output vector will be orthogonal to the input object's plane of symmetry. ∎

## Appendix C. Ablation Study

We further analyze OAVNN by conducting an ablation study. In particular, we investigate the two methods for learning the direction of symmetry: the symmetry detecting features and the orientation aware complex linear layer. First, we consider running a left-right segmentation experiment using only the symmetry detecting features. In particular, we run the planar symmetry detection algorithm on various left-right symmetric point clouds centered at the origin. We calculate the plane, which is centered at the origin and perpendicular to the average cross vector. We use this plane to divide each object into a left and right half-space. The results for various object classes are in Figure 7. On airplane objects, we obtain about 85% accuracy. Note that out of the four example classes shown in Figure 7,

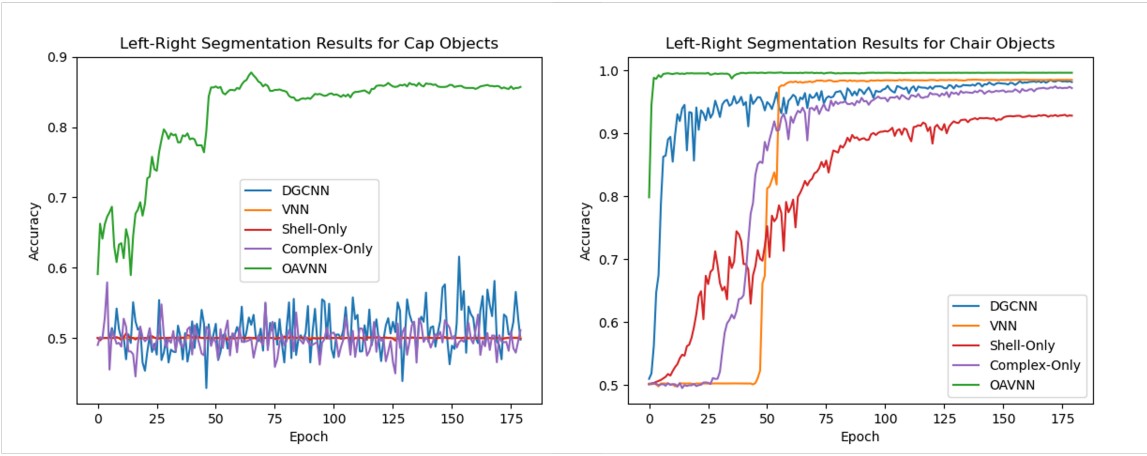

Figure 8: Results for the left-right segmentation experiment run on cap and chair objects from the Shapenet dataset (Chang et al., 2015). The solid line shows testing accuracies for five models: 1) DGCNN, a conventional segmentation network which is not O(3) equivariant (blue), 2) VNN an O(3) equivariant network that is ambiguous to symmetries (orange), 3) Shell-Only network which consists of only the symmetry-detecting features and the attention mechanism (red), 4) Complex-Only network which consists of only the orientation aware complex linear layer and the attention mechanism (purple), and 5) OAVNN which consists of the symmetry-detecting features, the orientation aware complex linear layer, and the attention mechanism (green). We see that of all the models, OAVNN obtains accurate results the quickest.

the table class does noticeably worse. This highlights one of the weaknesses of the planar symmetry detection algorithm. This algorithm will only work for objects with exactly one plane of symmetry. For objects like tables with two or more planes of symmetry, the algorithm outputs the zero vector. The algorithm also has suboptimal results for objects that are approximately symmetric about two planes (e.g. airplanes) or for point clouds with sampling or density issues.

Next, we create the Shell-Only network with only the symmetry detecting features and the attention mechanism. In particular, this network does not use the orientation aware complex linear layer. This network is similar to the VNN, however, it takes in two additional features: the symmetry detection features and the output of the attention mechanism. The results for this model are in red in Figure 3.

We also create the Complex-Only network with only the orientation aware complex linear layer and the attention mechanism. This network has the same architecture as the OAVNN, except the symmetry detecting features are not added. In other words, this network does not use the planar symmetry detection algorithm. The results of this network are in purple in Figure 3.

For airplane objects, only the VNN cannot complete the segmentation task. Therefore any network which can detect the direction of symmetry can complete the left-right

|  | Airplane | Cap | Chair |
|:---:|:---:|:---:|:---:|
| **DGCNN** | 0.901 | 0.496 | 0.981 |
| **VNN** | 0.501 | 0.500 | 0.985 |
| **Shell-Only** | 0.889 | 0.499 | 0.928 |
| **Complex-Only** | 0.876 | 0.511 | 0.972 |
| **OAVNN** | **0.905** | **0.857** | **0.996** |

Table 1: Results for the left-right segmentation experiment on airplane, cap, and chair objects from the Shapenet dataset (Chang et al., 2015). The table shows the average testing accuracies across three runs of five different models each trained for 200 epochs. The OAVNN model obtains the best final testing accuracy.

segmentation task for objects with one plane of symmetry. The DGCNN network takes roughly about 190 epochs to learn an accurate segmentation, and the Complex-Only network takes about 150 epochs. Surprisingly, the Complex-Only network (which is rotation equivariant) only does slightly better than the DGCNN (which is not rotation equivariant). The Shell-Only network takes about 75 epochs to learn an accurate segmentation.

The OAVNN model learns an accurate segmentation faster than all other models. This shows that when the symmetry detecting features, the complex linear layer, and the attention mechanism are combined, the model can complete the segmentation task the fastest. Additionally, as shown in Table 1, after training for 200 epochs the OAVNN model obtains the best final testing accuracy. Thus, both the planar symmetry detection algorithm and the complex linear layer are required to robustly learn the direction of symmetry and complete the symmetric segmentation task.

In Figure 8, we show results of the left-right segmentation experiment run on cap and chair objects. In both of these cases we again see that the OAVNN model is able to obtain an accurate segmentation the fastest. However, we do see two interesting behaviors.

First off, only the OAVNN successfully segments the cap objects. All other models cannot segment the cap objects even after 200 epochs of training. We expect all networks that are not ambiguous to symmetries, or all networks other than the VNN, to eventually complete the task. Unlike airplane and chair objects, the models may have a harder time learning how to segment cap objects because there are fewer cap objects in the dataset.

Second, the VNN model accurately segments chair objects after about 50 epochs of training. We would expect the VNN to be unable to complete the task because the network is ambiguous to symmetric objects like chairs. In these experiments, the VNN's behavior could be because the chair objects are not truly symmetric due to sampling issues. Additionally, it is likely that this behavior is more clearly seen with chair objects because airplane objects are symmetric about the left-right plane and approximately symmetric about the up-down plane. On the other hand, chairs are only symmetric about the left-right plane and are not approximately symmetric about any other plane. Therefore, the task of segmenting a chair is most likely easier than the task of segmenting an airplane.

