# OpenReview forum: "Breaking the Symmetry: Resolving Symmetry Ambiguities in Equivariant Neural Networks"
_NeurIPS.cc/2022/Workshop/NeurReps — NeurReps 2022 Poster_

### Official Review · Reviewer_QYER · 2022-10-11
**Review of "Breaking the Symmetry: Resolving Symmetry Ambiguities in Equivariant Neural Networks"**

**Confidence:** 4
**Soundness:** 3
**Presentation:** 3
**Contribution:** 2
**Overall Rating:** 5

**Summary:**

This paper presents a rotationally equivariant vector neural network architecture that enables tasks sensitive to reflectional symmetry, demonstrated on the toy task of left-right segmentation.


**Questions:**


Proofreading:

Use parenthetical citations (\citep{}) whenever the authors’ names are not part of the sentence.

Use journal/conference proceedings citations rather than arxiv.

Some commas are used incorrectly. For example:
* “The VNN’s linear layer is structured as, f_{lin}(X) = WX, where…” -> “The VNN’s linear layer is structured as f_{lin}(X) = WX, where…”
* “However, the DGCNN model Wang et al. (2018), a conventional segmentation network which is not equivariant takes a long time to learn.” -> “However, the DGCNN model (Wang et al., 2018), a conventional segmentation network which is not equivariant, takes a long time to learn.”

In Table 1, include standard deviations.



**Limitations:**

The authors present their algorithm for detecting reflectional symmetry, used within the OAVNN architecture, in appendices A.1. and B, explicitly noting the limitation that this algorithm can only handle point clouds with at most one plane of reflectional symmetry. The authors have not discussed if or how their architecture extends beyond O(3) and this specific decomposition into rotational and reflectional symmetries, as discussed above.


**Recommended Decision:**

2: Borderline

**Relevance:**

3: Solid fit

**Strengths And Weaknesses:**

To my knowledge, this work is sufficiently novel, albeit incremental in its current form. Although not completely necessary, I would like to see conceptual and empirical comparisons to the following unpublished work, at least in an appendix.
* Finkelshtein, B., Baskin, C., Maron, H., & Dym, N. (2022). A simple and universal rotation equivariant point-cloud network. arXiv preprint arXiv:2203.01216.

The authors build up the motivating problem in Section 3.2 very clearly, although this derivation could be moved to an appendix to save space. Definitions like 3.1 could also be moved, so Section 3 could be condensed to a single subsection to yield more space for discussing results.

The paper is clearly explained overall. The toy task of left-right segmentation clearly shows the benefit of OAVNN over some other approaches, including ablations, although more experiments towards more interesting and complex tasks would be greatly helpful to show the practical benefits of this method.

Similarly, the significance could be further motivated, such as at least giving some examples of tasks that “depend on input symmetry” beyond this toy problem of left-right segmentation? Can a more general theory and architecture be devised that allows the user to specify which symmetries of 3D point clouds are respected within the task vs. constrained as equivariance, or perhaps this division could even be learned as an extension of this work? These latter questions would make this work more interesting for this community.


**Submission Track:**

Extended Abstract (4 Page)

---

### Official Review · Reviewer_MvdT · 2022-10-14
**Breaking single planar symmetries in 3D: good introductory work**

**Confidence:** 4
**Soundness:** 4
**Presentation:** 3
**Contribution:** 3
**Overall Rating:** 7

**Summary:**

Equivariant networks whose inputs have their own intrinsic symmetries are limited in their expressive power. In particular, the output of an equivariant network on an input with a particular symmetry, must also respect that same symmetry. This is restrictive on problems like left-right segmentation of 3D objects represented as point clouds, which may (e.g. in the case of airplanes) themselves have a left-right symmetry. This work identifies this issue and presents a modification of Vector Neuron Networks to alleviate the issue. They explicitly design a layer to detect a planar symmetry in the input, and compose this with an “orientation aware linear layer” and an attention layer. Compared to traditional equivariant and non-equivariant networks, their OAVNN model learns more quickly over the course of training.

**Questions:**

1. How exactly are the three components of the architecture combined — what are the input directions J to the orientation aware linear layer? Pseudocode in the appendix describing the end-to-end network would be very helpful here. Moreover, the paper says “We also encourage detection of the symmetry direction by creating an orientation aware complex linear layer” at the start of A.2, but hasn’t the symmetry direction already been detected by the previous layer? Why are both the symmetry detection layer and the orientation necessary?

2. Unfortunately, I do not understand what the orientation aware complex linear layer does or how it is motivated. Perhaps a more thorough discussion of VNNs or DiffusionNet would aid the reader in understanding this layer.

3. It would improve the flow of a full-length writeup to justify why the obvious symmetry detection method — discretizing planar symmetries and checking them one by one — is computationally inefficient. Also, wouldn’t this work for the case of multiple symmetry detection?

4. It might also be helpful to include examples of other intrinsic symmetries or use cases than point cloud segmentation; for example, are there tasks in computational chemistry where the automorphism group of the graph poses a barrier to learning the task equivariantly?

5. The desired output function breaks equivariance, but still in a rather structured way, hence the improved performance over DGCNN. Can you formalize what class of non-equivariant functions you hope to restrict to with your architecture?

**Limitations:**

It seems that the symmetry detection method only necessarily succeeds in the case of a single-self symmetry. A 3D plus sign, for instance, has two symmetries, but a linear combination of them would be captured. Moreover, this methodology does not work for data that is not embedded in three dimensions or for other symmetry groups.

**Recommended Decision:**

3: Accept

**Relevance:**

4: Highly relevant

**Strengths And Weaknesses:**

Originality:
1. As far as I am aware, the architecture and contextualization of intrinsic symmetries for 3D object segmentation are novel. However, this is not the first work to consider symmetry detection in the context of equivariant networks: the authors should perhaps cite, and ideally compare their work with, “Finding symmetry-breaking Order Parameters with Euclidean Neural Networks” by Smidt, Geiger and Miller 2020.

Quality:
1. The submission seems to be technically sound.

Clarity:
1. The problem is well-motivated and clearly described, particularly with the aid of diagrams. The paper is also well-written in general. The discussion of experiments is clear and reasonably thorough. The biggest lack of clarity is in the description of the architecture, even in the appendix. Although the individual figures are helpful, more background and pseudocode would aid the reader in understanding how each layer type is motivated and integrated with the others.


Significance:
1. This work brings to light an interesting conundrum with equivariant architectures that has not been very thoroughly explored in the past. The experiments also compare to a reasonable set of baselines, and the ablation experiments are thorough.

2. As noted in limitations, the symmetry detection method is limited: it seems that it can only detect single planar symmetries. However, this seems acceptable for a preliminary work.

3. Figure 3, the main figure in the extended abstract, demonstrates that the OAVNN architecture learns more quickly (with respect to training epochs) than other architectures. However, I am not sure that this is a very meaningful measure of architecture quality. Unless there is a significant differential in computational resources between them, it seems that a comparison of ultimate accuracy with respect to the number of training samples would be more appropriate. (Indeed, this is the usual metric by which equivariant and non-equivariant architectures are compared.) I would expect to see a trade-off in which the non-equivariant architectures require more samples to reach the same accuracy, whereas the equivariant architectures are fundamentally over-constrained and cannot.


**Submission Track:**

Extended Abstract (4 Page)

---

### Official Review · Reviewer_HvMQ · 2022-10-15
**Equivariant network on data with symmetries**

**Confidence:** 4
**Soundness:** 2
**Presentation:** 3
**Contribution:** 4
**Overall Rating:** 8

**Summary:**

The extended abstract presents a rotation equivariant network on planar symmetric objects. It presents an attention mechanism that is aware of planar symmetry as well as an orientation-aware linear layer. The method is evaluated on the left and right segmentation on planar symmetric class planes in Shapenet dataset. The results are compared against other existing methods.

**Questions:**

1. Is there a connection to a canonical pose like in the 3D Point capsules paper? Do you think the orientation can be learned in a hierarchical fashion especially if this were to be extended in a mutli-planar symmetric case?

2. Are there experiments conducted on a toy dataset where the symmetries are not exact or slightly off-planar? Could you share the results?

**Limitations:**

As mentioned by the authors the planar symmetry algorithm fails in presence of multi-planar symmetries. In practice, there are cases with non-exact symmetries, especially considering the sampling issues hence learning only an exact symmetry may not necessarily outperform other methods with non-explicit symmetry learning.

**Recommended Decision:**

3: Accept

**Relevance:**

3: Solid fit

**Strengths And Weaknesses:**


OAVNN learns symmetry quickly and thus performs better compared to other models. The planar detection algorithm is intuitive and the use of the resultant orientation-aware layer is well-motivated. Overall a good addition to learning symmetry in self-symmetric objects with a focus on segmentation tasks. The extended abstract shows extensive writing and experimentation on their idea and shows good promise.

It would be helpful to have the actual OAVNN method in the extended abstract and with necessary additional details in Appendix, but the method was entirely in the Appendix sections.

**Submission Track:**

Extended Abstract (4 Page)

---

### Decision · Program_Chairs · 2022-10-21

Accept (Poster)